# Horse Welfare During Equine Chorionic Gonadotropin (eCG) Production

**DOI:** 10.3390/ani9121053

**Published:** 2019-12-01

**Authors:** Xavier Manteca Vilanova, Nancy De Briyne, Bonnie Beaver, Patricia V. Turner

**Affiliations:** 1School of Veterinary Science, Universitat Autònoma de Barcelona, 08193 Barcelona, Spain; xavier.manteca@uab.es; 2Federation of Veterinarians of Europe, 12B-1040 Brussels, Belgium; nancy@fve.org; 3Department of Small Animal Clinical Sciences, Texas A&M University, College Station, TX 77843, USA; bbeaver@cvm.tamu.edu; 4Charles River, Wilmington, MA 01887, USA; 5Department of Pathobiology, University of Guelph, Guelph, ON N1G 2W1, Canada

**Keywords:** horse welfare, PMSG, equine chorionic gonadotropin, pregnant mare

## Abstract

**Simple Summary:**

Equine chorionic gonadotropin or eCG is an important hormone produced by the placenta of pregnant mares and extracted from the blood of these same mares. This hormone is commonly used to enhance reproduction of pigs, dairy cows, sheep, beef cows, and goats. There are currently no alternative sources of this hormone. Horse welfare problems may arise if too much blood is collected at one time or during repeated collections or if the mares are not managed well. In some countries, mares are aborted several months into the pregnancy to improve efficiency since this permits them to become pregnant a second time in one year. We discuss approaches to protect the welfare of pregnant mares kept for eCG production.

**Abstract:**

Collection of blood from pregnant mares for extraction of equine chorionic gonadotropin (eCG) is a critical but relatively unknown and poorly regulated practice in the countries in which it occurs. Equine chorionic gonadotropin is a hormone that is widely used to enhance reproductive performance and management of dairy and beef cattle, sheep, goats, and pigs kept under intensive housing systems. eCG is extracted from the blood of brood mares between days 40–120 of gestation. Although alternatives have been sought, there is currently no efficacious replacement, natural or synthetic, for eCG. Recently, several animal welfare organizations have voiced concerns over the condition and treatment of pregnant mares kept for eCG production in some countries. Animal welfare issues may arise if mares are bled too frequently or if too much blood is collected at any time. In addition, these mares tend to be managed extensively on pastures with minimal veterinary oversight and they may be poorly desensitized and habituated to handling and other practices. This can lead to serious injuries and even death when mares are brought in for bleeding. This paper reviews the process of blood collection for eCG extraction and provides recommendations for ensuring mare welfare.

## 1. Introduction

Horses have been domesticated and raised for human use and enjoyment for thousands of years [1]. In addition to their frequent use for transportation and riding, horses are raised for food, agricultural assistance, companionship, sport, entertainment, education, competition, research, blood collection for serum or hormone extraction, and breeding. In all of these uses of the horse potential animal welfare issues have been identified (see, for example, [2]); however, the collection of blood from pregnant mares to produce eCG has received significant recent attention from animal welfare organizations and governments in the European Union (EU). Concerns have been raised regarding how horses are kept, handling methods and habituation practices for handling, volume of blood collected for eCG extraction, routine abortion of fetuses from pregnant mares to enhance productivity, and inadequate veterinary care [3]. Production of eCG is tightly linked to reproductive management of a number of food animal species, such as pigs, beef cattle, goats, and sheep, a multi-billion dollar global industry. The aim of this paper is to review the main horse welfare issues related to the production of eCG, to suggest strategies to make the process acceptable, and to encourage companies collecting brood mare blood for eCG production to carefully consider their ethical responsibility for these animals.

## 2. Overview of Equine Chorionic Gonadotropin Production

Equine chorionic gonadotropin (eCG), also known as pregnant mare serum gonadotropin (PMSG) or equine luteinizing hormone, is a glycoprotein hormone secreted by fetal-origin trophoblastic epithelial cells that form the endometrial cups [4,5]. These trophoblastic cells invade the endometrium around day 36–38 of gestation and begin to produce eCG approximately two days later, with peak production between days 55 and 70 of gestation. Production of equine chorionic gonadotropin continues until about day 110 (a range of 100–140 days) of gestation, at which point, the cells are targeted for destruction by an overwhelming maternal cellular immune response. It is thought that the function of eCG in the mare is to promote development of accessory corpora lutea, which helps to support the developing fetus [5]. To extract and purify eCG, pregnant mares are housed at production sites (‘blood farms’), small volumes of blood are withdrawn and tested by enzyme-linked immunosorbent assay (ELISA) for eCG content as the mares approach day 40 of gestation, and once confirmed positive, large volumes of blood are collected weekly or more often between gestation days 40 and 120 [4,5].

A number of mare and fetal factors determine the amount of eCG produced. For example, size of the mare is an important determinant, with larger horses producing more eCG, regardless of the size of the fetus being carried [6] and various genetic factors that are determined by the mare and the sire are also important (reviewed by [7]). Mare parity has a significant effect on eCG production, in that mean peak eCG concentration fell an average of 38% between the 3rd and 5th pregnancies for a given mare [8]. Similarly, research has suggested that both nonexercised and moderately conditioned mares produce more eCG than exercised or well-conditioned mares [8].

There are no international or industry guidelines or recommendations for blood collection from pregnant mares and the amount collected and the schedule of collection are highly variable between farms. In addition, some farms choose to abort the mares after day 90 of gestation, since pregnancy is not needed at that point for continued eCG production. This allows farms to rebreed mares such that two eCG production cycles are achieved each year, instead of one. On farms in which foals are raised, foals may be sold or sent for slaughter, depending on the country and breed of horse being used for eCG production.

## 3. Uses of Equine Chorionic Gonadotropin and Alternatives

In species other than horses, eCG induces the release of both luteinizing hormone (LH) and follicle stimulating hormone (FSH) (see [9] for a review of the functions of these hormones). Thus, use of eCG in animal production is widespread in countries with temperate climates, where natural estrus cycles of livestock would normally only support seasonal pregnancies. Uses of eCG include reversal of anestrus, induction of puberty, enhancement of fertility, and superovulation [7]. The hormone is used extensively, together with human chorionic gonadotropin to induce follicular growth and ovulation in immature gilts [10]. It is also used to induce ovulation in anestrus beef and dairy cattle, goats, sheep, and pigs as well as superovulation in the same species, for embryo transfer programs (reviewed by [11,12]). eCG is also commonly used to synchronize estrus on pig farms to allow more efficient management of pigs and employees and to reduce biosecurity issues associated with partial movement of animals in and out of barns [13]. Finally, eCG is used extensively in artificial insemination protocols in dairy cattle, in which it is thought to have beneficial effects on the developing embryo (reviewed by [12]).

Prolonged use of eCG is not without problems, as efficacy may wane over time following host production of an eCG neutralizing antibody [14]. Alternative methods to hormone use exist, including optimizing animal nutrition, photoperiod, and maturity of animals. However, these methods do not optimize reproductive performance of livestock to the same extent as eCG, making them unlikely to be implemented in modern intensive farming systems. Attempts have been made to use other hormone s alone or in combination to replace eCG use, but these studies have not resulted in an efficacious product [15]. Recent discovery of kisspeptin, a neuropeptide critical for central control of reproduction, has suggested another means of controlling estrus and ovulation (reviewed by [16]). A synthetic analog of kisspeptin has been demonstrated to have promising efficacy in synchronizing ovulations in goats [17]. More research is needed to optimize the molecule and evaluate its effects in other livestock species. At present, there is no efficacious synthetic alternative to eCG.

## 4. Regulation of Equine Chorionic Gonadotropin Production

Collection of blood from pregnant mares for production of eCG is difficult to track, but is thought to occur predominantly in Uruguay, Argentina, Iceland, Russia, Mongolia, and China. Within the EU and North America, there are no specific regulations concerning the welfare of live animals from which blood is collected, nor any animal welfare requirements for importing products manufactured outside of these areas [18]. Similarly, the World Organization for Animal Health (OIE) does not specify any regulations or oversight for this practice. EU legislation on animal by-products, defines a list of ‘safe’ countries to source blood from combined with legislation to prevent inadvertent transmission of specific animal diseases (e.g., blood must be collected from ‘live equidae’ under veterinary supervision at an approved establishment, mares should not show signs of notifiable diseases, blood should be inactivated as a preventative for certain diseases) [19,20]. The Uruguay Ministry of Agriculture published a new regulation for blood collection from horses and an accompanying manual of good animal welfare practices in 2017. However, this regulation is non-binding and only provides recommendations [21]. Companies collecting blood from brood mares for eCG extraction have not created any voluntary industry standards, thus, there is very little oversight of this practice.

## 5. Animal Welfare Issues Related to the Production of eCG

There are a wide range of conditions reported for housing and caring for these brood mares ranging from acceptable to clearly unacceptable, and each facility must be judged on its own merits. The lack of formal regulatory inspections for this industry in most countries in which it is currently occurring has meant that reports of animal neglect and abuse are anecdotal. Several graphic videos have been posted to social media by animal protection groups, but the timing and conditions under which these videos were collected are largely unknown.

### 5.1. Animal Welfare Issues Related to the General Husbandry and Care of Horses

Horse welfare should always be a priority, regardless of the purpose of keeping the animals. The large number of mares used by the eCG industry is likely to result in workers and stock people viewing the horses as a collective, rather than as individual animals. If horses are not observed carefully and regularly, animal neglect and abuse are more likely. There are numerous references available to describe welfare requirements for horses (see, for example, [22,23,24,25,26,27,28]).

An important factor influencing the welfare of mares used in the eCG industry is their environment. Because of the large number of animals associated with many facilities, in many countries, most of the mares are kept outdoors year-round on extensive pastureland. These pastures are poorly maintained and often consist of mixed features, including steppes, scrubland, and forests that are unsuitable for other types of farming, with scattered trees providing the only shelter. For these eCG horses, adverse weather may be an important factor affecting their welfare, especially as pregnancy proceeds and mares become less nimble navigating rough terrain. At some facilities, the pregnant mares are housed indoors, sometimes maintained under very crowded and intensive conditions. For these horses, welfare will be highly dependent upon the quality of their management, including air quality and temperature; bedding type, amount, quality; lighting; social interaction with conspecifics and choice in their environment; and schedules for feeding, watering, and exercise [27].

How horses are handled and habituated to handling will significantly impact horse welfare [27,28]. Many of the workers on these farms are unskilled, poorly educated laborers with minimal specific training and knowledge about horse behavior and stockmanship. There are anecdotal reports in a few countries of abuse of these brood mares, such as by beating, prodding, dragging fallen animals, and applying electric shocks to the anogenital areas of mares [29,30,31,32]. In some videos, brood mares are seen to fall and sustain secondary injuries following mistreatment, some of which require euthanasia. In these cases, appropriate and knowledgeable on-farm supervision is often lacking, and industry guidelines are largely ignored.

Horses must receive regular preventive care and veterinary oversight to maintain good health and well-being. This includes attention to hoof care, routine vaccinations, management of teeth, attention to hair coat and general body condition, as well as parasiticide treatment [27,28]. These measures are particularly important for pregnant animals and when large volumes of blood are being withdrawn on a regular basis. A veterinarian knowledgeable about equine care and well-being should be available at all times to evaluate sick or injured animals and to assist with end of life decision-making and euthanasia [27]. This care may not be available for all eCG facilities and anecdotal reports from some countries suggest that weak, emaciated, and lame horses are often present in a herd, and infected wounds are common [29,30,31,32]. On some farms, the lack of care results in estimated losses of between 25% and 30% of brood mares each year [29,30,31,32].

### 5.2. Animal Welfare Issues Related to Blood Collection

Once eCG levels are determined to be satisfactory, large groups of mares are herded into the holding facility where their blood is collected. In some cases, the horses may not be well socialized to humans and may be fearful in their presence, particularly when stock persons are poorly trained in preferred methods of working with horses and when they are rushed because of the large volume of blood collections to be done each day. On other farms, the mares live in close contact with the stock persons and are generally well habituated to handling. For blood collection, mares are individually herded into restraint chutes, a halter is fitted, and this is generally used to restrain the horses for bleeding. The quality of design of the restraint chutes is important in terms of reducing fear in horses. The presence of unfamiliar people, a solid front to the chute, and negative previous experiences elicit fear-based behaviors in horses, such as balking and attempts at escape. Inexperienced stock people may respond with physical abuse or shouting, further frightening the horses. Large bore needles are used to ensure rapid blood collection. On some farms, it is unclear whether the best practices are followed to ensure that: (i) bleeding sites are cleansed prior to venipuncture; (ii) sterile needles are used and discarded after each use; (iii) collection tubes are thoroughly cleaned between animals; and (iv) adequate hemostasis occurs to ensure that painful hematomas are avoided. Blood is harvested once or twice a week, depending on the facility and the country, throughout the approximate 80 days of eCG secretion [33].

Once the blood is collected, the mares are released back to the pasture where they graze without special dietary supplementation. Depending on the bleeding schedules, some horses can become weak, emaciated, and sick because of a compromised immune system [29,30,31,32]. Some of these mares will die in the pasture without further veterinary care [32]. Again, there are a wide range of conditions on farms and body condition may not be monitored closely.

### 5.3. Animal Welfare Issues Related to Abortion

Mares in poor body condition may undergo spontaneous abortions, but because the endometrial cups maintain production of eCG after day 90 and up to day 120, these mares can continue to be bled. Once eCg is no longer detectable, bleeding stops until the mare becomes pregnant again. After gestation day 130 day, some facilities allow mares to carry the foal to term and then sell the foals. On farms in Latin America, it is common for the pregnancy to be terminated so that the mare can be rebred for a second blood collection period that year [29,30]. Abortion is accomplished by injection of abortifacients, such as prostaglandins or by manually forcing open the cervix and rupturing the fetal membranes [29,30]. Either technique results in physical signs of parturition, with remains of aborted fetuses commonly found in the pastures. While it is known that mares produce less eCG after their third pregnancy, it is unknown how long the average brood mare remains within any given facility nor how culling and euthanasia decisions are made.

### 5.4. Animal Welfare Issues Related to the Transport of Horses (Not Only Mares, but Also Foals and Possibly Stallions)

Transporting horses within the eCG industry incurs the same welfare issues as for the horse industry as a whole (for example, see [34,35,36,37]). Transporting eCG horses occurs when the mare is first brought to the farm. Because the mares are usually pastured near the collection facility, they are herded rather than mechanically transported. Foals are generally not present in most facilities, and where pregnancies are allowed to go to term, the foal may be sold within the first 30–45 days. Mares that can no longer be used for blood collection may be shipped for slaughter. While this would normally not be more stressful than being transported on the farm, these mares may be excessively weak, making them more susceptible to falls and injury.

## 6. Horse Welfare Improvement Strategies for eCG Production

There are a number of strategies that may be used to ensure that pregnant mares used for blood collection for eCG are managed in ways that respect their welfare (summarized in Table 1).

### 6.1. Animal Welfare Strategies Related to the General Husbandry and Care of Horses

As discussed, most facilities that produce eCG maintain horses in groups under extensive pasture conditions, with permanent access to pasture throughout the year. In some cases, though, mares are moved into pens or box stalls during the period of blood production. Group housing and access to pasture has several advantages from an animal welfare standpoint, as horses can express normal social and foraging behaviors and benefit from exercise [27,28]. There are, however, several welfare problems that may arise in extensive pasture systems, including hunger and malnutrition due to poor foraging conditions or excessive stocking density, lack of drinking water, risk of predation, interanimal aggression, and lack of direct daily observation of animals that may be injured or experiencing other adverse conditions.

Adequate stocking density for horses kept on pasture depend on forage conditions. In properly managed temperate pastures, 0.5–1 hectare per horse has been recommended [38]. This, however, must be modified according to local forage conditions, which may well vary seasonally. When horses are moved into pens, care should be taken to ensure that all horses have access to feed and water and have enough space to move freely and lie down at the same time [27]. In wet and muddy conditions, horses should have access to a well-drained area to rest and they should have access to a run-in shelter to escape heavy winds, rain, snow or adverse weather conditions. Shelter space must be sufficient to accommodate all animals at the same time [27]. Supplementary feed (e.g., good quality hay) and mineral blocks must be provided when forage conditions are poor. It is recommended that forage is widely scattered to avoid competition between animals [27,38]. Horses must always have safe and palatable drinking water available, water troughs or other watering devices must be cleaned regularly [27].

Predation by feral dogs or, in some places, wild carnivores, may be a welfare risk, particularly for young foals. Livestock guarding dogs can be used to protect livestock (including horses) from predators. Livestock guarding dogs have been shown to be effective in a wide variety of environments and with different predator species, particularly when they are used together with other husbandry practices that help reducing the risk of predation. Husbandry practices that ensure the welfare and performance of livestock guarding dogs have been reviewed elsewhere [39].

Horses are social animals and keeping them in groups is likely to have positive effects on welfare. However, care should be taken to select groups that are compatible, and aggressive horses should be kept separated from the rest on a short term basis while the root cause of the behavior is assessed, evaluated for any physical or health issues, and then managed accordingly. If horses are individually kept in boxes, welfare problems may arise due to lack of social contact and the impossibility to express normal foraging behavior. Therefore, all stabled horses (excluding those animals that must rest for veterinary reasons) should benefit from a daily period in the field to allow them to graze and interact with other horses. This may reduce the risk of developing repetitive behaviors, such as cribbing, weaving, wind sucking, etc., which are all indicators of poor welfare [40,41].

Hot iron branding is used in several facilities to individually identify horses. However, hot branding is very painful and, when it is not a legal requirement, other methods of individual identification systems should be used [42]. Freeze branding is an alternative to iron branding, as it is less painful. However, freeze branding is more time consuming and requires more equipment than iron branding. Additionally, there are several safety issues to be considered when using freeze branding and, therefore, adequate training of personnel is required. Radio frequency identification (RFID) microchip placement is also a reliable means of identifying individual horses.

As for any other livestock production system, welfare assessment protocols are necessary to identify problems and monitor improvement strategies. Animal welfare must be assessed using a combination of several indicators. One assessment system that can be very useful is the Welfare Quality^®^ protocol, which is based on four principles of animal welfare, namely good housing, good feeding, good health, and appropriate behavior [43]. Each of these four principles comprises several criteria, with an overall total of 12 criteria, and each criterion is assessed through several indicators [43]. Although originally developed for cattle, pigs, and chickens, the Welfare Quality^®^ protocol has been adapted for use in horses.

Frequent direct or indirect (e.g., by remote camera or drone) observation of all horses by skilled staff is an important welfare requirement. Those working with or around animals need to be skilled in understanding and assessing horse behavior. Poor stockmanship causes chronic fear and is one of the main welfare problems in all farm species. Finally, it is essential that all horses on any farm collecting blood for eCG extraction have regular access to veterinary care, including routine health evaluations, preventative care (e.g., vaccinations, parasiticide treatment, dental prophylaxis, etc.), and emergency care, as needed.

### 6.2. Animal Welfare Improving Strategies Related to Bleeding

Regular bleeding of mares is an essential part of eCG production and may cause several welfare problems, including pain and stress due to the bleeding procedure as well as problems caused by the extraction of an excessive amount of blood (or by bleeding too frequently). Health and body condition of mares must be checked before bleeding and mares that are too thin or injured, or that show any signs of disease, must not be bled and should receive adequate treatment. Body condition scores for horses have been described and should be employed [44].

Bleeding must be done by trained, experienced personnel. The recommended bleeding procedure to be used in horses has been described elsewhere [45]. Blood should be collected aseptically using a new needle for each venipuncture and hemostasis should be applied after bleeding. There are no international blood collection standards for horses; however, using conservative blood volume estimates for non-sport horses [46], mares have 6.1 (draft) to 7.7 (Thoroughbred) L of blood per 100 kg BW. Using general research guidelines for safe blood collection in mammals [47], conservatively, this suggests that 6.1 L of blood can be safely collected every two weeks from a 1000 kg mare without compromising physiology. These figures may be modified if fluid replacement therapy is implemented. As the maximum volume of blood that can be extracted depends on body weight, mares should be weighed before bleeding. Alternatively, a conservative amount of blood based on the body weight of the lightest mares in the group should be extracted. If a mare is being bled routinely, the red blood cell packed volume (PCV) or the hemoglobin concentration should be checked regularly to determine when blood collection should be suspended so that the mare can recover from potential anemia.

A potentially interesting alternative to bleeding is plasmapheresis, i.e. the separation of plasma from blood cells so that only plasma is extracted from the animal, blood cells being immediately returned into the animal’s body. Although plasmapheresis may be advantageous in terms of animal welfare, care should be taken not to extract an excessive amount of plasma and hygiene and extraction practices would have to be improved to ensure that blood cells remained sterile following initial collection. Close supervision of mares and provision of drinking water immediately after bleeding or plasmapheresis is recommended.

Proper handling of mares during bleeding is essential to avoid or reduce stress. It may be useful to apply positive reinforcement techniques so that mares are less fearful of bleeding. Mares that are agitated or frightened should not be bled. As the response of an animal to handling is affected by its temperament, every effort should be made to use docile mares to produce eCG. Factors affecting horse temperament as well as methods to assess it have been reviewed elsewhere [41].

### 6.3. Animal Welfare Improvement Strategies Related to Abortion

Abortion of mares is not needed to produce eCG and, when it is done, its only purpose is to increase the amount of eCG produced per mare and year. Based on ethical grounds and considering that abortion is likely to be distressing and possibly painful for the animals, it is our opinion that it should not be performed, unless prescribed by a veterinarian and solely for therapeutic reasons.

### 6.4. Animal Welfare Improvement Strategies Related to the Transport of Horses

Transport of horses (including foals in many cases) is an unavoidable aspect of eCG production. Transport can compromise the welfare of horses as several stressors impinge upon the animals during a relatively short period of time. Recommendations to reduce transport stress in horses have been reviewed elsewhere (for example, see [27,36,48]).

Any horse that is weak, unable to move independently without pain or to walk unassisted, those with severe open wounds or prolapse are not fit for transport [27]. Loading and unloading are the most stressful parts of transport, particularly for young animals. Well designed and maintained ramps, as well as careful handling, are essential to reduce stress during loading and unloading.

Other aspects of transportation that must be considered on these farms include ensuring an appropriate stocking density, careful observations when mixing unfamiliar horses, the potential benefits of arranging horses to face the rear end of the vehicle, ensuring appropriate ambient conditions inside the trailer, steadiness of driving and vehicle movement [37], and transportation duration [27,48].

## 7. Conclusions

At this time, production of eCG is considered essential for appropriate reproductive management of many important livestock species and there is no efficacious synthetic replacement available. Collection of blood from pregnant mares for the purpose of extracting eCG has the potential to significantly compromise brood mare health and welfare. Consistent implementation of a range of strategies, as discussed, is needed to protect these mares. Strengthened national or regional inspection processes would help to assure mare welfare. In addition, those corporations and industries producing, selling, and using eCG have an ethical responsibility to ensure that practices for oversight, care, collection of blood, and management of pregnant mares are conducted humanely. More transparency of the industry is needed with an opportunity for open discussion between all stakeholders. Production of and strict adherence to industry guidelines that promote the welfare of horses being bled for eCG production would be an important first step in protecting these vulnerable animals.

## Figures and Tables

**Table 1 animals-09-01053-t001:** Summary of horse welfare improvement strategies during eCG production.

Potential Welfare Problem	Preventative Strategies
Chronic hunger	Maintain a recommended stocking density (0.5 to 1 hectare/horse in well managed, temperate pastures) and modify according to forage conditions Provide supplementary feed when forage conditions are poor
Chronic thirst	Ensure that fresh, potable water is always available
Physical discomfort	Ensure access to well-drained areas and shelter in wet, cold, and muddy conditions
Predation	Use livestock guarding dogs together with regular observation of animals in areas where predation occurs
Social stress	Ensure groups of mares are socially compatible, ensure animals have sufficient space and shelters for numbers held Aggressive horses—evaluate root causes and address medical, physical or management issues underlying behavior
Behavioral restriction (horses confined to box stalls)	Ensure that all stabled horses (excluding those animals that must rest for veterinary reasons) are offered a daily period in the field to allow them to graze, exercise, and interact with other horses
Pain caused by individual identification procedures	Avoid hot-iron branding, use alternative identification procedures such as freeze branding or micro-chipping
Stress caused by bleeding	Handle mares gently, ensure that they are habituated to handling and restraint methods, use positive reinforcement whenever possible. Do not bleed mares that are agitated or frightened Train all personnel handling mares in low-stress handling and restraint techniques
Negative consequences of removal of excessive amount of blood Negative consequences of removal of excessive amount of blood	Evaluate health and body condition of mares before bleeding, weigh animals to ensure blood collection volumes are appropriate Do not bleed sick mares or mares in poor body condition; ensure that these mares have access to veterinary care, if needed Limit blood collection to a maximum of 6 L/1000 kg BW every 2 weeks, replace blood with fluids, if possible
Pain and discomfort caused by abortion	Do not induce abortion
Stress caused by transport	Follow internationally accepted guidelines for horse transportation

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
