# Peer review of "Horse Welfare During Equine Chorionic Gonadotropin (eCG) Production"

_animals, 2019, doi:10.3390/ani9121053_

Round 1

Reviewer 1 Report

19 it might be worth including the list of countries in this section as I was interested in this starightaway

25 repetition of 'reproductive'

32 use of habituation refers to a particular learning theory - perhaps suggest using the word 'training' instead so as to avoid inferring one particular method needs to be used (desensitisation would be more appropriate in the case of these mares)

57 'begin' to produce

64 ELISA - in full then abbreviate

101 hormones

114 OIE in full

161 re-word (grammar)

219 I am concerned about advice given regarding 'aggressive' horses - how would this be judged, and also aggression in horses is often a sign of pain, is it appropriate to separate aggressive horses anyway?

249 as above

258 minor spelling

260 RFID in full

264 reference for Welfare Quality protocol in sentence it first appears in

Author Response

Thank-you for your comments to improve this MS. We have addressed them below and in the revised MS.

19 - add countries - we are concerned that we may inadvertently exclude some countries that are aborting mares as it is a very secretive industry, so it is preferable to list the concern without being too specific.

25 - revised, as suggested

32 - thank-you for your comment. We did actually mean 'habituation' as it should be practiced. We have also added desensitization.

57 - corrected, as suggested

64 - defined, as suggested

101 - correct in current text (singular form)

114 - defined, as suggested

161 - revised, as suggested

219 - we have addressed by recommending assessment of the root cause(s) with appropriate medical, physical or management changes 

249 - please see above

260 - defined, as suggested

264 - added, as suggested

Reviewer 2 Report

The authors describe the current state of the equine chorionic gonadotropin (eCG or pregnant mare serum gonadotropin) collection industry, with a focus on welfare of mares used in the industry. The review is well-written article and was easy to read. However, it was disappointing that the manuscript contributes little to advance our knowledge about this industry. All of the information presented in the article is readily accessible by reviewing a couple of on-line articles; thus, the importance of this manuscript is fairly low.

Next, it would have be of much greater value if the authors could have provided more specific details about the scope of the industry. For example, an estimate of the number of mares actually involved in the industry, perhaps subdivided by country, would be helpful. Do the farms/ranches that are in the industry have to be registered or licensed in any of the countries? Similarly, what data may be available about the sale of eCG products? Knowledge of sales might allow estimation of the number of mares used in the industry.

Finally, a major oversight of recommendations for improved mare welfare provided in the manuscript is ignoring the role of the responsibility of the companies that manufacture and sell eCG. If the companies involved would insist on minimum standards of equine welfare, and back up these standards by inspection of the farms/ranches on a regular basis, equine welfare would improve. In fact, the companies selling the product are the only “regulatory agency” with any ability to ensure mare welfare. The authors are encouraged to review and consider making a comparison to the pregnant mare urine (PMU) farms in Canada. The PMU an industry that is well-supervised by Pfizer, the pharmaceutical company that extracts estrogens/progestogens from mare urine to make several pharmaceutical products used both in human and veterinary medicine. PMU ranches have annual contracts with Pfizer and if animal welfare is substandard these contracts can be cancelled. Corporate responsibility and oversight of the eCG industry will be challenging with multiple countries involved, but it is likely the only link in the chain from mare’s blood to thin final product that can lead to improved equine welfare in the industry. This should be emphasized by the authors of this manuscript. A first step could be a recommendation for a multinational industry meeting to establish minimal guidelines for equine welfare in the eCG industry, followed by steps to ensure implementation of welfare oversight. Until the companies assume their responsibility, the industry (and mare welfare) is unlikely to change.  

Specific changes

Line 57 – change “beginning” to “begin”

Line 172 – remove “to” after “may”

Line 195 – delete “t”

Lines 221-222 – replace “extensive conditions” – not sure what this means

Line 226 – same concern – replace “extensive”

Line 283 – change “scales” to “scores”

Line 316 – consider changing “must” to “should”

Please ensure that all references are accurate

Author Response

We thank the reviewer for their thoughtful comments. As the reviewer is aware, information concerning numbers of farms, numbers of horses, etc, is not publicly available. We agree that it would be preferable to publish numbers by countries, but these farms are not required to register and report specific numbers of mares used for eCG production or volumes of blood collected in any country. Horses are maintained and bled for multiple reasons for human/veterinary medical product development  (e.g., production of reagents for Coombs testing, serum extraction for licensed passive immunization products, such as snake anti-venom, tetanus toxoid, hemagglutination inhibition assays for influenza virus, etc., isolation of estrogens for medications for post-menopausal women, and so on), thus, it is extremely difficult to parse out the eCG-specific farms or horses.

The practice of maintaining brood mares on blood farms for eCG harvesting is largely unknown to most veterinarians, let alone the public, at large. This paper was written on behalf of the World Veterinary Association's Animal Welfare Working Group to begin to address this specific issue. The authors' desire in publishing this paper is to raise the profile of the practice into a more public arena and to begin a dialogue with those producing and purchasing eCG for use for livestock reproductive management. 

It is true that Pfizer maintains horse farms in Manitoba, Canada for blood collection; however, this is for a different purpose - explicitly for the production of FDA-licensed medicines for human use. These farms are subject to more rigorous inspection because of this. We appreciate the reviewer's suggestion of strengthening recommendations for corporate responsibility and have added this to the general conclusion of the paper.

Line 57 - revised, as suggested

Line 172 - deleted, as suggested

Line 195 - deleted, as suggested

Lines 221 and 226 - added 'pasture' - the point being made is that these animals are not maintained under intensive farming conditions, but rather under extensive pasture conditions (minimal direct food provided, minimal observations, etc)

Line 283 - revised, as suggested

Line 316 - revised, as suggested